

# Brief communication: The influence of mica-rich rocks on the shear strength of ice-filled discontinuities

Philipp Mamot[1], Samuel Weber[1], Maximilian Lanz[1], and Michael Krautblatter[1]

[1]Chair of Landslide Research, Technical University of Munich, Munich, Germany

**Correspondence:** Philipp Mamot (philipp.mamot@tum.de)

**Abstract.** A temperature- and stress-dependent failure criterion for ice-filled rock (limestone) joints was proposed in 2018 as an essential tool to assess and model the stability of degrading permafrost rock slopes. To test the applicability to other rocks, we now conducted experiments with mica schist/gneiss which provide the maximum expected deviation of lithological effects on the shear strength due to strong negative surface charges affecting the rock-ice interface. Retesting 120 samples from $-10$ to $-0.5°$ C and 100 to 400 kPa, we show that even for controversial rocks the failure criterion stays unaltered, suggesting that the failure criterion is transferable to mostly all rock types.

## 1   Introduction

The climate-related change of the thermal conditions in steep bedrock permafrost can lead to rock slope destabilisation or failure (e.g. Gruber and Haeberli, 2007) potentially triggering large scale hazards via process chains (Huggel et al., 2012). A number of failures in bedrock permafrost have exposed residual ice at their shear and detachment planes (Keuschnig et al., 2015; Phillips et al., 2017; Ravanel et al., 2017; Weber et al., 2018; Walter et al., 2020). These observations indicate the occurrence of ice-filled rock discontinuities and their importance as controlling factor for the stability of degrading permafrost rock slopes. While the shear strength of rock joints can be increased by ice fillings due to adhesion and rock-ice-interlocking (Gruber and Haeberli, 2007), warming can reduce the strength of ice-filled joints (Krautblatter et al., 2013). To improve attempts to accurately assess the stability of rock slopes with degrading permafrost due to climate change, we will have to better understand the effect of warming on the shear strength of ice-filled joints.

So far, the shear failure of ice-filled rock joints has been studied in a number of direct shear tests using samples of concrete and ice (Davies et al., 2000; Günzel, 2008). Mamot et al. (2018) added a series of constant strain shear tests with real rock samples. The ice-filled rock joints were represented by "sandwich"-like limestone-ice-limestone samples. Normal stresses of 100, 200 and 400 kPa were applied to the ice-filled discontinuities, which simulated rock overburdens of 4, 8 and 15 m, respectively. A brittle failure criterion was developed for these overburdens and validated for rock temperatures from $-8$ to $-0.5°$ C. The failure criterion is based on Mohr-Coulomb and contains a temperature- and stress-dependent cohesion and friction which decrease upon warming.

Mamot et al. (2018) postulate that their failure criterion can be applied to shear planes in all rock types, even though they solely used limestone for the laboratory experiments. The authors discussed potential influences of the rock type on the shear





strength and they emphasised the need to study a potential dependency in additional experiments. An inventory of rock slope failures in the central European Alps by Fischer et al. (2012) shows that the rock types, which are potentially involved in the fracture of ice-filled rock joints, are not only limestone but also gneiss and granite. Most of the detachment zones are likely affected by permafrost and roughly follow the assumed lower altitudinal boundary of permafrost. Rock slopes within

this boundary are expected to respond sensitively to warming and a related reduction in stability (Nötzli et al., 2010), and predominantly consist of gneiss, limestone or schist at altitudes from $2000 - 3000\,\mathrm{m}$ a.s.l. and mostly of gneiss or granite at altitudes $> 3000\,\mathrm{m}$ a.s.l. (Fischer et al., 2012).

Among the rocks observed to be involved in permafrost rock slope failures, gneiss and schist are metamorphic rocks with a pronounced foliation and typically show bands with a concentrated abundance of aligned, platy mica. These bands can form

weak zones where shear planes develop more easily (Shea and Kronenberg, 1993). Furthermore, basal cleavage surfaces of mica carry a strong negative surface charge compared to other minerals like quartz, feldspar or calcite. When in contact with water, this leads to the formation of a common but specifically strong electrical double layer (Fenter et al., 2000; Bourg and Sposito, 2011). Such an electrical double layer causes a homogenous alignment of at first water molecules, and, with freezing, structural integrity in ensuing ice crystals (Dosch et al., 1996). As a result, adhesion at the interface is increased and the

process of phase change upon warming is delayed. While the mean equilibrium freezing point in permafrost rocks is depressed to $-0.7 \pm 0.4°\,\mathrm{C}$ (Krautblatter, 2009), mica-rich rocks with strongly attractive surfaces are observed to increase the freezing temperature (Alba-Simionesco et al., 2006). Close below the melting point, this results in a crystalline contact layer on mica-rich surfaces while inner layers remain fluid. In contrast, a liquid layer forms along the rock-ice interface on weakly attractive silica-rich surfaces while the absorbed water in the mineral is crystalline (Alba-Simionesco et al., 2006). Consequently, we

theoretically expect a stronger adhesion at the rock-ice interface for mica-rich rocks, presumably leading to a higher shear strength close below $0°\,\mathrm{C}$.

Therefore, this study aims to verify if the failure criterion by Mamot et al. (2018) accounts for (i) the relevance of gneiss for permafrost rock slope failures, (ii) the abundance of gneiss and schist within the lower permafrost boundary and (iii) the potentially significant effects of foliated metamorphic rocks with high mica content on the shear strength of ice-filled joints.

Hence, this manuscript addresses the following question: Is the failure criterion for ice-filled rock joints by Mamot et al. (2018) valid for rocks with different mineral composition, specifically containing mica?

## 2 Methods

Two different rock types with a considerable mica content were selected for this study: gneiss that originates from the Matterhorn (45°58'52" N, 07°40'14" E, $3218\,\mathrm{m}$ a.s.l.), Switzerland, and mica schist that was involved in the Ramnanosi landslide,

close to the village of Flåm (60°49'41" N, 07°08'59" E, $750\,\mathrm{m}$ a.s.l.), Norway (Fig. 1a).

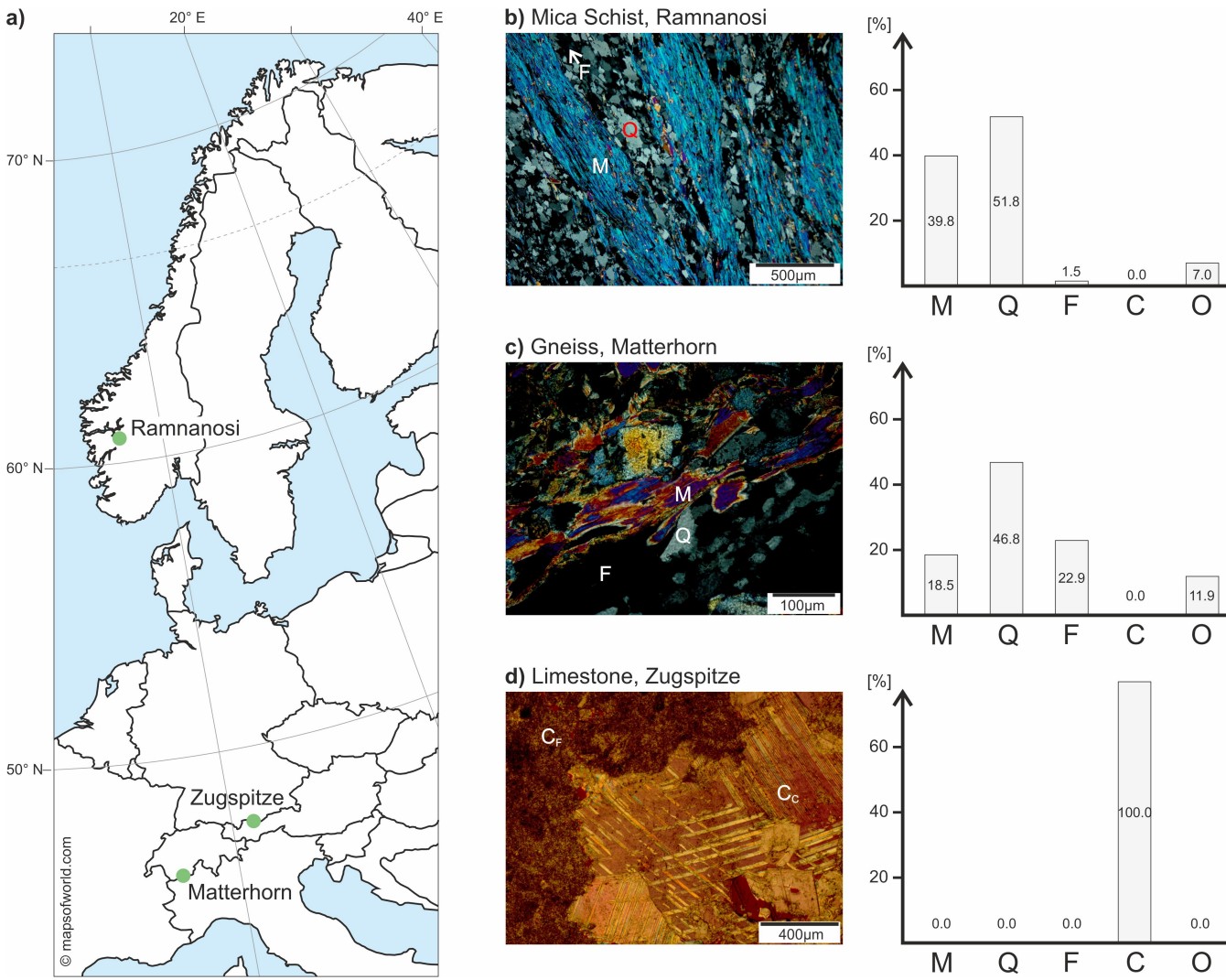

**Figure 1.** a) Map showing the locations where samples were collected. b), c) and d) Associated results of thin section analyses, where M refers to mica, Q to quartz, F to feldspar, C to calcite (Cf = fine grains; Cc = coarse grains) and O to others.





## 2.1 Petrographical analysis

A thin section analysis was conducted to determine the mineral composition and the amount of mica in the rock samples. Two thin sections were prepared of each rock type. To account for the anisotropic nature of both gneiss and mica schist, all thin sections were prepared from cuts perpendicular to the foliation of the rocks. The analysis was conducted through cross polarised light microscopy with an Olympus DP26 microscope which is standardised according to the DIN ISO 8576 (2002). The proportionate mineral compositions were recorded in a PELCON Automatic Point Counter due to point-counter analysis as recommended by Chayes (1949). On each thin section, two point-counter tracks of at least 100 points were indexed. We estimated the mica content on sample surfaces via image histogram analysis.

## 2.2 Shear experiments

The sample preparation, setup and procedure of the tests were conducted according to Mamot et al. (2018) to guarantee comparability: (i) Each rock–ice–rock sandwich sample consists of two piece of rock with a surface roughness of 80 grains per inch and its gap is filled with a $3.5 \pm 0.5\,\mathrm{mm}$ thin ice layer. (ii) We used the same custom built shear apparatus installed in a temperature-controlled cooling box. (iii) We applied a constant strain rate of $5 \times 10^{-3}\,\mathrm{s}^{-1}$ provoking brittle fracture of ice and thereby representing the final acceleration stage of a rock slope failure.

Tests were performed at temperatures of $-10, -6, -2, -1$ and $-0.5\,^\circ\mathrm{C}$ and at normal stress levels of $100, 200$ and $400\,\mathrm{kPa}$. Four tests were conducted for each rock type and each combination of temperature and normal stress, leading to a total of 120 tests.

## 3 Results and interpretation

### 3.1 Mineral composition relevant for rock-ice interfaces

The mica schist is a fine-grained rock with well-developed foliation. Layers of dark mica, predominantly biotite, and light quartz and plagioclase alternate in the rock. In the thin section analysis (Fig. 1b) the mica schist shows very distinct bands of mica (39.8%) in a ground mass mostly comprised of quartz (51.8%). Other minerals identified in the sample are feldspar and chlorite as well as possibly sillimanite and amphibole. An average porosity of 1.3% was determined in the thin section analysis. The gneiss is clearly laminated and shows bands of mica (18.5%), yet it is less distinctly layered than the mica schist (Fig. 1c). The main components are quartz (46.8%) and feldspar (22.9%), other minerals identified in the sample are chloritoid and epidote as well as traces of chlorite, amphibole and rutile. Furthermore, the porosity of the gneiss is considerably low (0.9%, Draebing and Krautblatter, 2012) compared to the mica schist. Both rock types show a high abundance of mica minerals at the sample surfaces presumably leading to a stronger polarity than at limestone surfaces. The latter are only constituted of calcite without any occurrence of mica (Fig. 1d).





## 3.2 Shear tests of rock-ice interfaces with mica-rich rocks


Mamot et al. (2018) subdivided the paths of shear stress and shear strain into five distinct stages including (i) consolidation, (ii) adjustment to the sample holder, (iii) buildup of the shear stress, (iv) failure and (v) post-failure behaviour. The same pattern can be identified by the presented experiments with mica-rich rocks. We also observe a general decrease in peak shear stress with increasing temperature from $-10$ to $-0.5°\,$C at all tested normal stress levels, without any systematic difference between

the samples of mica schist and gneiss (Fig. 2). Overall, the measured peak shear stresses of this study lie well within or close around the range of the failure criterion (dark blue area in Fig. 2). Even the experiments conducted at $-10°\,$C fit mostly well within the expected values of the same failure criterion, although they are outside of its proposed valid temperature range. Nevertheless, for the same temperature and at a normal stress of $400\,$kPa, the measured peak shear stresses tend to fall below the failure criterion. This pattern is also visible in the previous tests with limestone (Fig. A1) and in rock-ice shear experiments

by Günzel (2008), possibly due to the beginning transition from brittle to ductile failure with higher rock overburden leading to a lower shear strength (Renshaw and Schulson, 2001). When approaching the melting point, above $-2°\,$C, the measured peak shear stresses slightly exceed the calculated range of the failure criterion.

Figure 3 illustrates the temperature-dependent proportion of failure types that were observed in the shear tests for distinguishing rocks with and without mica content. Failure within the ice is very dominant for mica-rich rocks in the temperature

range from $-10$ to $-1°\,$C, but absent at $-0.5°\,$C. In contrast, rock without mica (limestone) shows a gradual decrease in the proportion of failure within the ice and a gradual increase in the proportion of failure along the rock-ice interface with warming. Overall, it is remarkable that the fracture along the rock-ice interface and mixed fracture are the only failure types at $-0.5°\,$C for limestone and mica-rich rocks.

## 4 Discussion

In Mamot et al. (2018) the question remained open if the failure criterion for ice-filled rock joints was transferable to other rock types than limestone. As Fischer et al. (2012) showed that gneiss and schist are among the most relevant rocks involved in permafrost rock slope failures, this study aims at testing the applicability of the failure criterion to these rocks. Furthermore, gneiss and schist were selected as they are characterised by foliation with high mica content, which may potentially affect the shear strength of ice-filled joints close below $0°\,$C.

We quantified the shear strength of ice-filled discontinuities in mica-rich rocks which fits well to the temperature- and normal stress-dependent failure criterion (Fig. 2), thereby indicating its validity for mica-rich rocks. It is remarkable that the failure criterion is rather conservative for conditions close to $0°\,$C compared to the shear tests with mica-rich rocks. However, this underestimation of shear strength does not state a problem for any purpose of rock slope stability assessment and, hence, does not diminish the applicability of the failure criterion. The higher shear strength close below $0°\,$C for mica-rich rocks has been

expected by the authors and may be explained by the higher adhesion at the rock-ice interface, the delayed phase change upon warming and the absence of a liquid layer along mica-rich joint surfaces and close to the melting point (see Sect. 1). The stabilising effect of the rock-ice-contact by mica becomes also evident when looking at the temperature-dependent distribution

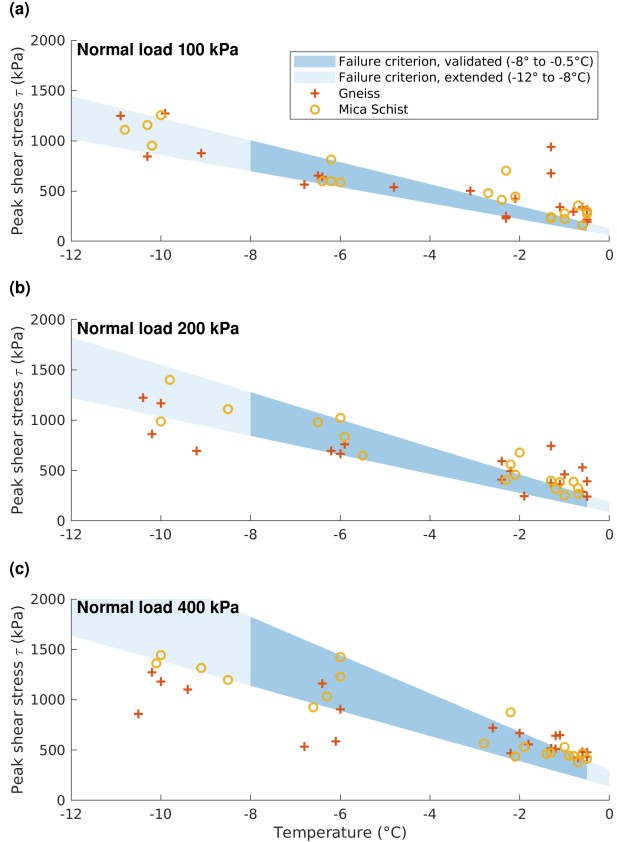

**Figure 2.** Peak shear strength across sub-zero temperature of ice-filled rock joints constituted of gneiss (red crosses) and mica schist (orange circles) for normal stresses of a) $100\,\text{kPa}$, b) $200\,\text{kPa}$ and c) $400\,\text{kPa}$. The validated range of the failure criterion by Mamot et al. (2018) is marked in dark blue while an extended section down to $-12°\,\text{C}$ is displayed in light blue.

of failure types (Fig. 3): While in mica-rich rocks failures along the rock-ice contact only dominate above $-1°\,\text{C}$, in limestone (without mica) the same failure type gradually gains importance upon warming and above $-6°\,\text{C}$.

Previous publications on the shear strength of ice-filled rock joints relate to a small number of tests on samples with ice and concrete, generally not below $-5°\,\text{C}$ (Günzel, 2008; Davies et al., 2000). Krautblatter et al. (2013) developed a first failure criterion which based on the experiments by Günzel (2008). Five years later, Mamot et al. (2018) proposed an improved failure criterion which refers to rock-ice-rock samples and covers a broader range of bedrock temperatures. This study demonstrates that the failure criterion by Mamot et al. (2018) is surprisingly resilient as it can be applied to (i) different failure types including

the fracture in ice and along the rock-ice contact, (ii) a wide range of temperatures relevant for bedrock permafrost ($-0.5$ to $-8°\,\text{C}$), (iii) a wide range of relevant stress conditions ($100 - 400\,\text{kPa}$) and (iv) mostly all rock types relevant for permafrost rock slope failures, as the metamorphic mica-rich rocks tested in this study represent the expected maximum deviation of potential lithological effects on the shear strength of ice-filled rock joints. This strong deviation is established due to three





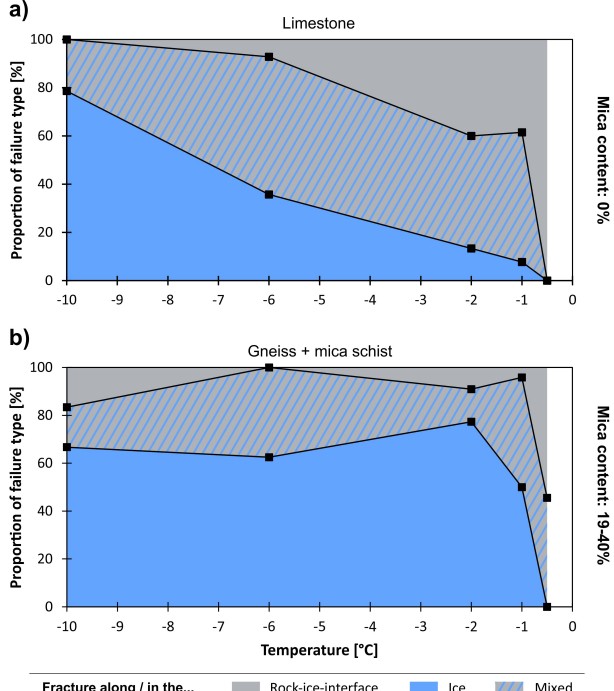

**Figure 3.** Proportions of failure types versus temperature for a) mica-free (0%) and b) mica-rich (19-40%) surfaces in ice-filled rock joints. Failure in mica-rich joints is dominated by fracturing inside the ice layer, whereas at temperatures warmer than -1°C failure is controlled by fracturing along the rock-ice interface and the mixed type. However, in mica-free joints the proportion of ice fracturing decreases gradually when approaching the melting point while rock-ice fracturing gains importance, especially at temperatures warmer than -1°C.

characteristics of the tested rocks: (i) They are foliated and have typically a high amount of mica aligned parallel to the shear surfaces. (ii) The platy and parallelly aligned mica grains lead to a very low surface roughness potentially reducing the shear strength. (iii) The strong negative surface charge results in an elevated adhesion and equilibrium freezing point presumably leading to a higher peak shear strength.

## 5  Conclusions

In this study, we performed 120 constant strain rate shear tests on ice-filled joints in gneiss and mica schist to investigate a potential influence of metamorphic foliated rocks with high amount of mica on the shear resistance of ice-filled discontinuities. Based on the experiments, we could demonstrate a systematic increase of peak shear strength at temperatures close to $0°C$, which is most likely caused by the existence of mica. However, overall our data lie well within the failure criterion for ice-filled rock joints by Mamot et al. (2018). As the tested mica-rich rocks represent the expected maximum deviation of potential lithological effects on the shear strength, we conclude that the failure criterion is transferable to mostly all rock types relevant for permafrost rock slope failures.





*Data availability.* All data which refer to the test conditions and samples, as well as the measured shear stress values, are provided in the Supplements in a *.xlsx file.

## Appendix A

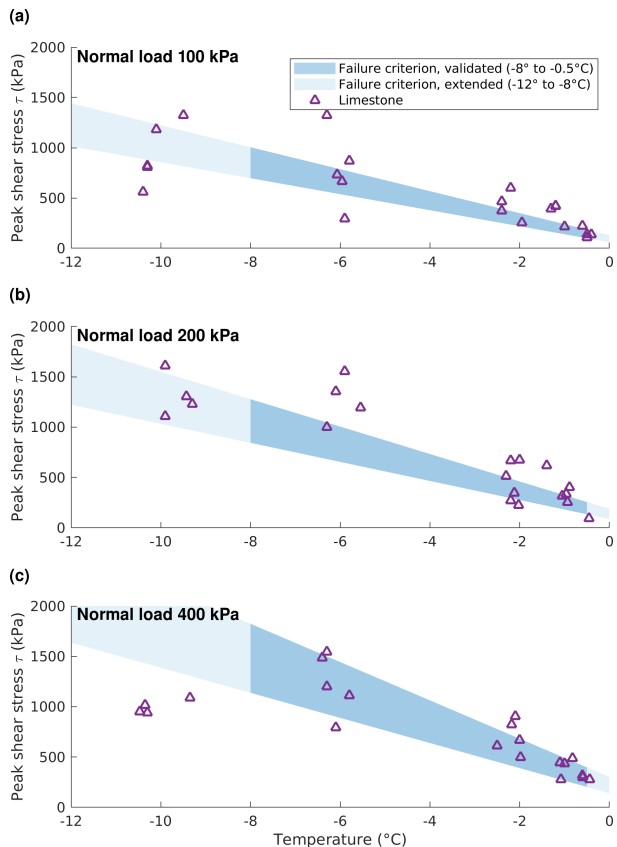

**Figure A1.** Peak shear strength versus sub-zero temperature for ice-filled rock joints constituted of limestone. The presented data are taken from previous tests by Mamot et al. (2018). Relationships are plotted for normal stresses of a) $100\,\mathrm{kPa}$, b) $200\,\mathrm{kPa}$ and c) $400\,\mathrm{kPa}$. The validated range of the failure criterion by Mamot et al. (2018) is marked in dark blue while an extended section down to $-12°\,\mathrm{C}$ is displayed in light blue.

*Author contributions.* Philipp Mamot (PM), Samuel Weber (SW) and Michael Krautblatter (MK) designed the shear experiments. Max Lanz (ML) prepared the samples and performed the tests, as well as the thin section analyses. He was supervised by PM and MK. Analysis of the data was conducted by SW, ML and PM. The manuscript was written by PM, SW and ML, with substantial contribution of MK.






*Competing interests.* The authors declare that they have no conflict of interest.

*Acknowledgements.* We gratefully acknowledge Andreas Grovan Aspaas for providing, coring and cutting the rock samples from Norway.
Furthermore, we thank Cordula Bode for preparing the thin sections used for the petrographic analysis. Finally, we thank the Fritz und Lotte
Schmidtler-Foundation financing the TUFF fellowship that is held by Samuel Weber.



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
