# Peer review of "Brief communication: The influence of mica-rich rocks on the shear strength of ice-filled discontinuities"

_The Cryosphere, 2020_

## Referee Comment (RC1) · Anonymous Referee #1 · 6 Mar 2020

General comments:

A well written Brief communication that complements the earlier work (Mamot et al 2018) very well.

1. Does the paper address relevant scientific questions within the scope of TC? yes

2. Does the paper present novel concepts, ideas, tools, or data? Yes, new data

3. Are substantial conclusions reached? Not substantial, but a valuable contribution to the understanding of behaviour of ice filled rock joints.

4. Are the scientific methods and assumptions valid and clearly outlined? yes

5. Are the results sufficient to support the interpretations and conclusions? Some additional comments are requested for points (i), (ii), and (iii) of the discussion (please see below)

6. Is the description of experiments and calculations sufficiently complete and precise to allow their reproduction by fellow scientists (traceability of results)? Yes, but some additional comments are requested on the sample preparation (see below).

7. Do the authors give proper credit to related work and clearly indicate their own new/original contribution? yes

8. Does the title clearly reflect the contents of the paper? yes

9. Does the abstract provide a concise and complete summary? yes

10. Is the overall presentation well structured and clear? yes

11. Is the language fluent and precise? yes

12. Are mathematical formulae, symbols, abbreviations, and units correctly defined and used? yes

13. Should any parts of the paper (text, formulae, figures, tables) be clarified, reduced, combined, or eliminated? I recommend combination of Figure 2 and Figure A1.

14. Are the number and quality of references appropriate? yes

15. Is the amount and quality of supplementary material appropriate? yes

Specific comments:

Line 55 / Figure 1: also give the coordinates of the Zugspitze location

Line 65 sample preparation: Please comment on the alignment of mica parallel to the surfaces of the samples used in the experiments: presumably the samples were cut parallel to the foliation. How does the mica content on the sample surface compare with the mica content in the thin sections? Were the samples cut through mica-rich

bands (weak bands and therefore more likely to form fractures in a rock mass)? How many samples were prepared and what was the variation of these samples in terms of mica content?

Line 68-69: a strain rate is compared with an acceleration. Please make sure that you compare like for like.

Line 83: I suggest to replace "stronger polarity" with "higher concentration of negative surface charges".

Line 94: replace "rock-ice" with "concrete-ice"

Figure 2 and Figure A1: I recommend to include the limestone data points from Figure A1 into Figure 2, as this makes it easier for the reader to directly compare the data. I may become necessary to increase the size of Figure 2.

Figure 3: please add a comment on the reliability of the data: how was the failure type observed? Can you give an error estimate for the failure type identification?

Lines 129-132: Point (i): Please comment on the alignment of mica parallel to the surfaces of the samples used in the experiments: natural rock fractures form along mica platelets that are not perfectly parallel. A cut rock surface therefore will expose cuts through a mica grain rather than the surface of the silica sheet. Can you give an estimate how the surface charges of a cut surface differ from the surface charges of a natural fracture?

Point (ii): I agree with the statement; however, in your experiments you use surfaces with the same roughness. Can you comment on the effect of the different surface roughness on the shear strength of natural joints in limestone vs. joints in gneiss or mica schist?

Point (iii): I suggest to replace "presumably" with "possibly". Please comment to what extent the reduction of shear strength from (ii) and the increase of shear strength from (iii) cancel each other out.

Line 136: I suggest to replace "systematic increase" with "slight increase". The highest points of the data clouds of the silica samples are higher than the highest points of the limestone samples; however, the data clouds overlap and about half of the limestone data points are also above the failure criterion.

———————————————————

---

## Referee Comment (RC2) · Lukas U. Arenson (Referee) · 11 Apr 2020

[referee-annotated manuscript omitted]

---

## Author Comment (AC1) · 16 Apr 2020

Dear referee, we thank you very much for your positive feedback and the constructive, valuable suggestions and comments related to the content and editorial issues. We carefully addressed all of them and listed the changes in detail in the *.pdf-file enclosed.

Best regards, Philipp Mamot, on behalf of all co-authors

Please also note the supplement to this comment:
https://www.the-cryosphere-discuss.net/tc-2020-18/tc-2020-18-AC1-supplement.zip

---

## Author Response (AR1)

**Review 1 (by anonymous referee)**

Dear referee,
we thank you very much for your interesting, constructive and valuable suggestions and comments related to the content and editorial issues. We carefully addressed all of them and listed the changes in detail hereafter.

Philipp Mamot, on behalf of all co-authors

RC = Referee comment
AR = Author response

Line 55/Figure 1:
**RC:** "Also give the coordinates of the Zugspitze location"
**AR:** As suggested by the referee, we provided the coordinates of the Zugspitze location in the text (lines 59-61 in the revised manuscript):
*"The limestone samples used for the precedent laboratory tests by Mamot et al. (2018), to which this study refers, were picked from the Zugspitze (47°25'21" N,10°59'13" E, 2900 m a.s.l.), Germany."*

Line 65:
**RC:** "Sample preparation: Please comment on the alignment of mica parallel to the surfaces of the samples used in the experiments: presumably the samples were cut parallel to the foliation. How does the mica content on the sample surface compare with the mica content in the thin sections? Were the samples cut through mica-rich bands (weak bands and therefore more likely to form fractures in a rock mass)? How many samples were prepared and what was the variation of these samples in terms of mica content?"
**AR:** We revised the information on the sample preparation and the mica content. A part of the text was shifted from Sect. 2.1 to the second paragraph of Sect. 2. in the following way (lines 62-69 in the revised manuscript):
*"A thin section analysis was added to the direct shear tests to determine the mineral composition and the amount of mica in the rock samples. The thin sections were taken from the same rock blocks from which the cylinders for the shear tests were cored. Two thin sections were prepared of each, the gneiss and the mica schist; the results were averaged per rock type. One thin section was produced of the limestone. To account for the anisotropic nature of both gneiss and mica schist, all samples (Sect. 2.1 and 2.2) were prepared from*

*cuts parallel to the foliation of the rocks. As such, we assume a similar mica content for both the thin sections and the sample surfaces in the shear tests.*

*[…]*

*2.1 Petrographical analysis*

*The thin section analysis was conducted through cross polarised light microscopy with an Olympus DP26 microscope..."*

As we only prepared two thin sections per rock type, we cannot give a variation in terms of the mica content.

However, we expanded the supplementary material (xlsx.-file) by the measured mineral compositions of the rock types used for the laboratory tests. This novelty was added to the section "Data availability" (lines 163-164 in the revised manuscript):

*"Data availability. All data which refer to the test conditions and samples, as well as the measured shear stress values and mineral compositions, are provided in the Supplements in a *.xlsx file."*

Line 68-69:

**RC:** "A strain rate is compared with an acceleration. Please make sure that you compare like for like."

**AR:** We changed the related sentence as follows (line 77-78 in the revised manuscript):

*"(iii) We applied a constant strain rate of $5 \times 10^{-3} s^{-1}$ provoking brittle fracture of ice and thereby representing the well-advanced stage of rock slope failure."*

Line 83:

**RC:** "I suggest to replace "stronger polarity" with "higher concentration of negative surface charges"."

**AR:** We modified the wording as proposed (line 94 in the revised manuscript).

Line 94:

**RC:** "Replace "rock-ice" with "concrete-ice"

**AR:** We exchanged the wording as proposed (line 105 in the revised manuscript).

Figure 2 and Figure A1:

**RC:** "I recommend to include the limestone data points from Figure A1 into Figure 2, as this makes it easier for the reader to directly compare the data. It may become necessary to increase the size of Figure 2."

**AR:** We combined both figures as suggested by the reviewer. For this, we also changed the caption of Fig. 2:

[Figure]

**(a)**

**(b)**

**(c)**

**Figure 2.** *Peak shear strength across sub-zero temperature of ice-filled rock joints constituted of gneiss (red crosses), mica schist (orange circles) and limestone (grey triangles). The relationships are plotted for normal stresses of a) 100 kPa, b) 200 kPa and c) 400 kPa. The limestone data are added from previous tests by Mamot et al. (2018). The validated range of the failure criterion by Mamot et al. (2018) is marked in dark blue while an extended section to −12 °C and to 0 °C is displayed in light blue.*

Further, we modified the in-text-link to Fig. A1 as follows (line 105 in the revised manuscript): *"This pattern is also visible in the previous tests with limestone (grey triangles in Fig. 2) and…"*

Figure 3:
**RC:** "Please add a comment on the reliability of the data: how was the failure type observed? Can you give an error estimate for the failure type identification?"

**AR:** We added a short comment on this in the methods Section 2.2 (line 81-83 in the revised manuscript):

*"As in the tests by Mamot et al. (2018), the type of failure was identified qualitatively by visual inspection of the failure surfaces immediately after removing them from the shear apparatus. Samples which did not allow a definite failure type classification were assigned to the mixed failure."*
* * *
Lines 129-132:

**RC:** "Point (i): Please comment on the alignment of mica parallel to the surfaces of the samples used in the experiments: natural rock fractures form along mica platelets that are not perfectly parallel. A cut rock surface therefore will expose cuts through a mica grain rather than the surface of the silica sheet. Can you give an estimate how the surface charges of a cut surface differ from the surface charges of a natural fracture?"

**AR:** The missing information was provided as follows (lines 141-143 in the revised manuscript):

*"(i) They are foliated and have typically a high amount of mica aligned subparallel within major shear planes. This property and the resulting effect of the surface charge are expected to be more emphasised along natural joints than along the tested surfaces, as these were cut within intact rock samples."*
* * *
Lines 129-132:

**RC:** "Point (ii): I agree with the statement; however, in your experiments you use surfaces with the same roughness. Can you comment on the effect of the different surface roughness on the shear strength of natural joints in limestone vs. joints in gneiss or mica schist?"

**AR:** We expanded Point (ii) by the following information on the effect of a different rock type-dependent surface roughness on the shear strength of natural joints (lines 144-149 in the revised manuscript):

*"(ii) The platy and subparallelly aligned mica grains lead to a very low surface roughness potentially reducing the shear strength. This effect will become more relevant at temperatures close to 0 °C where we observe a higher proportion of fractures along the rock-ice interface. As it is hard to define a representative surface roughness for typically diverse natural fractures, and to guarantee reproducibility of the laboratory rock surfaces, we standardised the joint surface roughness in our tests. Therefore, we assume the effect of varying surface roughness and its dependence on the rock type to be visible in natural fractures, but not in our tests."*
* * *
Lines 129-132:

**RC:** "Point (iii): I suggest to replace "presumably" with "possibly". Please comment to what

extent the reduction of shear strength from (ii) and the increase of shear strength from (iii) cancel each other out."

**AR:** We exchanged the wording from "presumably" to "likely" (line 150 in the revised manuscript).

Further, we added a short comment on how the reduction of shear strength from (ii) and the increase of shear strength from (iii) cancel each other out (lines 150-154 in the revised manuscript):

*"(iii) The strong negative surface charge results in an elevated adhesion and equilibrium freezing point which likely leads to a higher peak shear strength.*

*Due to the uniform surface roughness in the presented tests, we are not able to determine the extent to which the reduction in shear strength by a lower surface roughness (see ii) may offset the increase in shear strength by a strong negative surface charge (see iii). But, overall, we expect the observed mica-dependent higher shear strength close to 0 °C to be suppressed slightly."*

Line 136:

**RC:** "I suggest to replace "systematic increase" with "slight increase". The highest points of the data clouds of the silica samples are higher than the highest points of the limestone samples; however, the data clouds overlap and about half of the limestone data points are also above the failure criterion."

**AR:** We exchanged the wording as proposed by the referee (line 158 in the revised manuscript).

**Review 2 (by Lukas U. Arenson)**

Dear Lukas Arenson,
we thank you very much for your constructive and valuable suggestions and comments related to the content and editorial issues. We carefully addressed all of them and listed the changes in detail hereafter.

Philipp Mamot, on behalf of all co-authors

RC = Referee comment
AR = Author response

Lines 2-5:
**RC:** The referee suggested to
(1) write "rock types" instead of "rocks"
(2) delete "now"
(3) replace "experiments" with "laboratory tests"
(4) add a comma
(5) add "at temperatures" and "normal stress of"
**AR:** We adjusted the proposed changes in the revised manuscript as follows (lines 2-5 in the revised manuscript):
*"To test the applicability to other rock types, we conducted laboratory tests with mica schist/gneiss, which provide the maximum expected deviation of lithological effects on the shear strength due to strong negative surface charges affecting the rock-ice interface. Retesting 120 samples at temperatures from −10 to −0.5 °C and normal stress of 100 to 400 kPa,…"*

Line 8:
**RC:** The referee suggested to delete "the" and "of" and modify "change" to "changes".
**AR:** Adjusted as proposed (line 8 in the revised manuscript).

Line 22:
**RC:** The referee asked for a reference to the Mohr-Coulomb failure criterion.
**AR:** We added the reference in the text and included the two citations in the reference list (line 22 in the revised manuscript):

*"The failure criterion is based on Mohr-Coulomb (a combination of Coulomb (1776) and Mohr (1900)), and contains a temperature- and stress-dependent cohesion and friction which decrease upon warming."*

Line 25:
**RC:** The referee replaced "for" with "by" and "the" with "their".
**AR:** We changed the wording as proposed (line 28 in the revised manuscript).

Line 26:
**RC:** "rock type" was added, "in" was modified by "with", "laboratory tests" was proposed for "experiments".
**AR:** Adjusted as proposed (line 29 in the revised manuscript).

Lines 27-28:
**RC:** "You mean the rock types in which failure occurred? Sentence unclear."
**AR:** We revised the sentence to clarify what we aimed to say (lines 29-31 in the revised manuscript):
*"An inventory of rock slope failures in the central European Alps by Fischer et al. (2012) shows that the rock types, in which failure occurred and which potentially include the fracturing of ice-filled rock joints, are not only limestone but also gneiss and granite."*

Line 29:
**RC:** The referee suggested to add "for the area".
**AR:** Adjusted as proposed (line 33 in the revised manuscript).

Line 33:
**RC:** The referee suggested to add "types".
**AR:** Changed as proposed (line 36 in the revised manuscript).

Line 44:
**RC:** The reviewer suggested to exchange "we" by "one would".
**AR:** Adjusted as proposed (lines 47-48 in the revised manuscript).

Caption Fig. 1:
**RC:** "Use same fonts and subscript as in Figure."
**AR:** We revised the caption as followed:

*"…b), c) and d) Associated results of thin section analyses, where M refers to mica, Q to quartz, F to feldspar, C to calcite ($C_F$ = fine grains; $C_C$ = coarse grains) and O to others."*

Line 61-62:

**RC:** The reviewer replaced "due to" with "for a" and the syntax of a sentence was changed.

**AR:** We realised the proposed changes in the following way (lines 71-72 in the revised manuscript):

*"Two point-counter tracks of at least 100 points were indexed on each thin section. We estimated the mica content on sample surfaces via image histogram analysis."*

Line 66:

**RC:** The reviewer added an "s" at the end of "piece".

**AR:** Modified as proposed (line 75 in the revised manuscript).

Line 70:

**RC:** The reviewer suggested to add "respectively".

**AR:** Adjusted as proposed (line 80 in the revised manuscript).

Line 83:

**RC:** The reviewer suggested to exchange "are" with "is".

**AR:** Adjusted as proposed (line 95 in the revised manuscript).

Line 88:

**RC:** The referee reminded to set "laboratory tests" instead of "experiments"

**AR:** We exchanged the wording as proposed by the reviewer (line 99).

Lines 90-96:

**RC:** The referee suggested to
   (1) add a comma
   (2) write "close to" instead of "close around"
   (3) delete "well"
   (4) write "outside the valid temperature range proposed" instead of "outside of its proposed valid temperature range"
   (5) relocate a part of the sentence to the end of it
   (6) replace "visible" with "noted"
   (7) add "the"

(8) extend "close to the melting point" with "of the ice".

**AR:** Revised as proposed (lines 101-108 in the revised manuscript):

*"Overall, the measured peak shear stresses of this study lie well within, or close to the range of the failure criterion (dark blue area in Fig. 2). Even the laboratory tests conducted at −10 °C fit mostly within the expected values of the same failure criterion, although they are outside the valid temperature range proposed. Nevertheless, the measured peak shear stresses tend to fall below the failure criterion for the same temperature and at a normal stress of 400 kPa. This pattern is also noted in the previous tests with limestone (grey triangles in Fig. 2) and in the concrete-ice shear experiments by Günzel (2008), possibly due to the beginning transition from brittle to ductile failure with higher rock overburden leading to a lower shear strength (Renshaw and Schulson, 2001). When approaching the melting point of the ice, above −2 °C, the measured peak shear stresses slightly exceed the calculated range of the failure criterion."*

Line 98 and 101:

**RC:** "I suggest to use ratio instead of proportion (also in the Figure). To me ratio sounds more natural."

Further, the reviewer asked to add "of ice" behind " the melting point".

**AR:** Adjusted in the text (line 109 in the revised manuscript), in Fig. 3 and in the respective caption (see below), as proposed.

[Figure]

*Figure 3. Ratios of failure types versus temperature for a) mica-free (0 %) and b) mica-rich (19-40 %) surfaces in ice-filled rock joints. […] in mica-free joints the proportion of ice fracturing decreases gradually when approaching the melting point of ice…"*

Lines 105-109:

**RC:** The referee suggested to

1. relocate the word "other" behind "rock types"
2. write "demonstrated" instead of "showed"
3. replace "involved in rock slope failures within the Alpine mountain permafrost belt" with "involved in permafrost rock slope failures"
4. write "for those rock types" instead of "to these rocks"
5. exchange "as" by "since"
6. write "below" instead of "to".

**AR:** Adjusted as proposed (lines 116-120 in the revised manuscript).

Lines 110-117:

**RC:** The referee suggested to

(1) add a comma

(2) write "seems to be" instead of "is rather"

(3) exchange "below" by "to"

(4) extend "close to the melting point" with "of the ice"

(5) delete a reference to Section 1

(6) write "strengthening" instead of "stabilising".

**AR:** Revised as suggested by the referee (lines 121-127 in the revised manuscript).

Fig. 2:

**RC:** The reviewer asked to change the three sub-headings from "normal load" to "normal stress".

**AR:** Adjusted as proposed (see below):

**(a)**

[Figure]

**(b)**

[Figure]

**(c)**

[Figure]

Caption of Fig. 2:
**RC:** "But also an extension from -0.5 °C to 0 °C?"
**AR:** We added the information that the extended section also refers to temperatures between -0.5 and 0 °C:

*"…The validated range of the failure criterion by Mamot et al. (2018) is marked in dark blue while an extended section to −12 ◦C and to 0 ◦C is displayed in light blue."*
* * *
Lines 118-119:
**RC:** "Just be consistent to talk about interface, not contact"
Further, the referee suggested to change the sentence to "…gains in importance at temperatures warmer than -6 °C"
**AR:** Adjusted as proposed (lines 129-130 in the revised manuscript).
* * *
Line 127:
**RC:** The reviewer added "in the Alps".
**AR:** Adjusted as proposed (line 138 in the revised manuscript).
* * *
Caption of Fig. 3:
**RC:** The reviewer added "of ice" to "the melting point".
**AR:** Adjusted as proposed.
* * *
Line 129:
**RC:** The referee added "rock types".
**AR:** Adjusted as proposed (line 140 in the revised manuscript).
* * *
Line 132:
**RC:** "More out of curiosity and for reference to put this into perspective, what is the shear strength (or if not available he UCS) of the unfrozen rocks?"
**AR:** We provided two sentences on the shear strength of unfrozen joint surfaces to Section 1 (lines 23-26 in the revised manuscript). The presented unfrozen values by Krautblatter et al. (2013) are based on the same normal stresses (100-400 kPa) and on the same rock type (Wetterstein limestone) which were used in the laboratory tests by Mamot et al. (2018). The surface roughness of the unfrozen samples (grit of 24 grains per inch) differed only slightly from the one of the ice-filled samples (grit of 80 grains per inch). Shear strength values of

unfrozen gneiss or mica schist surfaces were not available and, hence, could not be presented. The new information in the article is as follows:

*"When warming from -1 or -0.5 °C leads to thawing and a subsequent loss of the ice infill, the shear strength of unfrozen joints reduces slightly by approximately 100 kPa (Krautblatter et al., 2013; Mamot et al., 2018). However, the unfrozen shear strength is 400-1000 kPa lower when compared with the one of ice-filled joints at temperatures between -2 and -10 °C."*

Lines 134-140:

**RC:** The referee proposed to
- (1) change "performed" into "carried out"
- (2) write "laboratory tests" instead of "experiments"
- (3) replace "of" by "in", "existence" by "presence" and "lie" by "correspond"
- (4) add "introduced by"
- (5) replace "mostly all rock types" by "wide variety of rock types"
- (6) add "in the Alps" at the end of the sentence

**AR:** Adjusted as proposed (lines 156-162 in the revised manuscript):

[revised manuscript text omitted]